# Concave Side of Proximal Thoracic Zone Vulnerable to Pedicle Screw Perforation in Adolescent Idiopathic Scoliosis Surgery: Comparative Analysis of Pre- and Intraoperative Computed Tomography Navigation

**DOI:** 10.3390/jcm14134729

**Published:** 2025-07-03

**Authors:** Tomohiro Yamada, Yu Yamato, Tomohiko Hasegawa, Go Yoshida, Tomohiro Banno, Hideyuki Arima, Shin Oe, Koichiro Ide, Kenta Kurosu, Yukihiro Matsuyama

**Affiliations:** Department of Orthopaedic Surgery, Hamamatsu University School of Medicine, 1-20-1 Handayama, Chuo-ku, Hamamatsu 431-3125, Japan; yamato@hama-med.ac.jp (Y.Y.); hasetomo@hama-med.ac.jp (T.H.); goy@hama-med.ac.jp (G.Y.); t.banno@hama-med.ac.jp (T.B.); arihidee@yahoo.co.jp (H.A.); mecersior@gmail.com (S.O.); 1de.5one60@gmail.com (K.I.); kenta_hamamatsu@yahoo.co.jp (K.K.); spine-yu@hama-med.ac.jp (Y.M.)

**Keywords:** pedicle screw, computed tomography navigation, adolescent idiopathic scoliosis

## Abstract

**Background:** The aim of this study was to assess pedicle screw (PS) accuracy and identify perforation patterns using computed tomography (CT) navigation in adolescent idiopathic scoliosis (AIS) surgery. **Methods:** A total of 107 AIS patients were retrospectively reviewed. Preoperative CT navigation was used in 48 patients (853 screws), and intraoperative CT with a second 3D scan was used in 59 patients (1059 screws). Postoperative CT images were analyzed using the Rampersaud grading system. **Results:** Overall PS accuracy (grade A + B) was significantly higher in the intraoperative CT group than the preoperative group (97% vs. 95%, *p* = 0.008). In Lenke type 1 cases, accuracy was also higher in the intraoperative group (97.8% vs. 95.1%, *p* = 0.014). Grade D perforations were most frequent on the concave side of the proximal thoracic (PT) zone in both groups. Ten screws were re-inserted during surgery in the intraoperative group based on findings from the second 3D scan. **Conclusions:** The concave PT zone is a common site for PS misplacement. Intraoperative CT navigation with a second 3D scan enhances PS accuracy compared with preoperative CT navigation.

## 1. Introduction

Pedicle screws (PSs) comprise the most prevalent segmental fixation system to stabilize all three columns of the spine in adolescent idiopathic scoliosis (AIS) surgery. After Suk et al. provided their initial results of thoracic PS placement [1], PSs have provided stable anchor points connected to rods, which can aid the surgical management of complex spinal deformities in AIS. However, PS perforation has the potential risk of correction loss, neurological deficits, and organ damage [2,3]. Recent morphometric analyses confirm that up to 30–35% of pedicles in AIS patients show anatomical abnormalities, particularly in the proximal thoracic region [4,5]. The visceral structures at risk during PS insertion in a scoliosis patient differ at each vertebral level. There is also a higher risk of injury to the spinal cord at the PT concave side screws at the T2 and T3 levels and the main concave sides at the T5–9 levels [6].

To improve insertion accuracy, several reports have revealed that various assistive methods, including intra- and preoperative computed tomography (CT) navigation, enable improved PS placement accuracy compared with the conventional freehand technique [7,8,9]. Zhang et al. recently reported that intraoperative navigation systems facilitated PS insertion in the apical region comparably with preoperative CT-based navigation [10]. Intraoperative CT navigation features advantages and disadvantages. First, surgeons do not require spine registration, which can cause errors in preoperative CT navigation [11]. Additionally, a surgeon could use a second three-dimensional (3D) scan to confirm screw position. In contrast, longer surgeries requiring multiple X-rays increase patient radiation exposure [12].

However, few studies have addressed the anatomic risk zones for pedicle screw misplacement and whether intraoperative re-evaluation using a second CT scan improves accuracy in a meaningful way [10,12]. We believe that determining the effectiveness of intraoperative CT navigation will allow surgeons to optimize the use of intraoperative CT navigation. Therefore, this study aimed to compare the PS accuracy and perforation characteristics between pre- and intraoperative CT navigation in AIS surgery. Moreover, it aimed to identify the specific zone where the PS perforation rate increased in both groups. Finally, it aimed to demonstrate the usefulness of the second intraoperative 3D CT scan that contributed to the improved accuracy rates.

## 2. Materials and Methods

This retrospective radiological study was based on a prospective patient database from a single academic spinal surgery department. Between 2013 and 2022, we conducted a retrospective review of patients with AIS who underwent corrective fusion surgery (T2–L5) via pedicular segmental instrumentation using intra- or preoperative CT navigation. The intraoperative spinal 3D imaging system was developed using an O-arm (Medtronic Navigation, Louisville, CO, USA). We regularly performed AIS surgery using intra- or preoperative CT navigation. Radiation dose reduction protocols were applied during intraoperative CT navigation. Scanner settings were adjusted to use the lowest acceptable dose for adequate image resolution, in line with pediatric radiation safety guidelines. All scans were performed using a low-dose protocol whenever feasible.

To compare screw accuracy under the strictest conditions, we compared PS accuracy using intra- or preoperative CT from same-day surgery based on Lenke AIS [13] and zone classification [14]. To maximize accuracy, axial scans, as well as sagittal and coronal reconstructions, were reviewed. This study was approved by our local institutional review board (#20-189), which waived the requirement for informed consent because of the retrospective nature of the research.

### 2.1. Patient Allocations

To reduce selection bias, patients were quasi-randomly allocated to either intraoperative or preoperative CT navigation based on the surgical schedule. On days when two AIS surgeries were planned, one patient was assigned to intraoperative CT navigation and the other to preoperative CT navigation. This approach ensured that both navigation methods were used under similar conditions by the same surgical team.

### 2.2. Surgical Technique

All surgeries were performed under general anesthesia with the patient in the prone position on an Allen table. Preoperative prone CT scans were used to identify the anatomic variation resulting from the rotation, angulation, and translation of each vertebral segment, as well as the extent of spinal canal abnormalities. Routine scans were performed to assess the thin pedicle, whereas extensive CT scan cuts were used for stereotactic guidance. When cancellous bone was lacking in the pedicle on the preoperative CT scan, we canceled the PS placement. During surgery, a reference frame was fixed to the spine close to the vertebrae to be instrumented [15]. In the preoperative CT navigation group, registration was performed at each of the three vertebrae from the caudal side. In the intraoperative CT group, each scan was performed to cover as many vertebrae as possible from the most caudal vertebra within the reference frame. In the intraoperative CT group, a second intraoperative 3D scan was routinely performed in all cases after pedicle screw placement to confirm accuracy and allow for immediate revision if necessary before rod placement.

To determine the PS insertion points, the facets were identified after the careful exposure of the involved spinal segments and the localizing pedicle probe was placed at the base of the superior articular facet, as well as the midpoint of the transverse process and the pars interarticularis. While referencing the point bar, we chose the insertion point and trajectory. During the intraoperative navigation surgery, we performed a second 3D scan to evaluate the screw direction; if unacceptable, we re-inserted it. Intraoperative screw re-insertion was performed for all screws with Rampersaud grade D perforations (>4 mm breach). For grade C breaches (2–4 mm), re-insertion was determined at the surgeon’s discretion based on intraoperative safety considerations.

### 2.3. Radiological Evaluation

Rampersaud et al.’s criteria were used to evaluate the screw position and direction of screw breach using a postoperative axial CT scan [16]. According to these criteria, patients were categorized into four grades: A, entirely in the pedicle; B, <2 mm breach; C, 2–4 mm breach; and D, >4 mm breach. Oversized screws touching the medial and lateral cortices were considered part of grade A unless the breach was >2 mm. Based on the fact that screws perforating the canal for up to 2 mm were considered acceptable [17,18], grades A and B were categorized as accurate.

We also evaluated intraoperative re-insertion cases in the intraoperative CT group. Operative time and blood loss were compared between groups, while radiation dosage using the dose length product (DLP) calculation was determined in the intraoperative CT group.

### 2.4. Statistical Analysis

Descriptive statistics are depicted as mean and standard deviation and were calculated for demographic data and radiographic parameters. Differences in individual and radiographic parameters were assessed using an unpaired *t-*test, a chi-squared test, and Fisher’s exact test. All statistical computations were performed using Statistical Package for the Social Sciences software (version 26.0; IBM Corp., Armonk, NY, USA). Values of *p* < 0.05 were considered statistically significant.

## 3. Result

A total of 1948 PSs (107 patients) were analyzed. Patient age, sex distribution, Lenke classification, and preoperative major Cobb angle did not significantly differ between groups (Table 1). The screw accuracy rates were 97% in the intraoperative CT group, significantly higher than in the preoperative CT group (95%). Table 2 shows the perforation rates (grade C or D) based on Lenke classification. The perforation rate was the highest among Lenke type 2 AIS cases of the preoperative CT group (7%), as well as Lenke type 3 AIS cases of the intraoperative CT group (11%). The perforation rate among Lenke type 1 AIS cases was higher in the preoperative (5%) versus intraoperative CT group (2%) (*p* = 0.014).

Table 3 shows the grade D perforation distribution according to zone classification in both groups. Grade D perforation occurred much more often in the preoperative group versus the intraoperative group in the concave proximal thoracic (PT) and transitional main thoracic/lumbar (MT/L) zones. Overall, there were 18 perforations in the preoperative CT group versus 13 perforations in the intraoperative CT group. In the intraoperative CT group, the second 3D scan enabled the re-insertion of the perforated screw. Screw re-insertion decreased the number of grade D perforations from 13 to 4. Grade D perforations occurred much more often in the preoperative group versus the intraoperative CT group in the concave PT and transitional MT/L zones.

Table 4 shows the grading shift in the perforated screw before and after the second 3D scan in the intraoperative CT group. A total of 10 PSs were re-inserted after the confirmational scan, leading to screw salvage. The mean operative time was significantly higher in the intraoperative than preoperative CT group (*p* = 0.033). However, there was no intergroup difference in estimated blood loss. The mean dose length product (DLP) in the intraoperative group was 365 mGy-cm.

### 3.1. Representative Case 1

A 14-year-old girl presenting with Lenke type 2A AIS and a major Cobb angle of 56° at T7–L1 underwent corrective fusion surgery at T2–L2 using preoperative CT navigation. Postoperative neurological testing revealed no deterioration; however, the postoperative CT showed a medial grade D perforation at the right T6 at the concave side of the PT zone (Figure 1). The patient was asymptomatic, but she was electively revised postoperatively to mitigate the risk of delayed complications.

### 3.2. Representative Case 2

A 14-year-old girl presenting with Lenke type 5C AIS and a major Cobb angle of 47° at T10–L3 underwent corrective fusion surgery at T6–L3 using intraoperative CT navigation. A second intraoperative 3D scan indicated a medial grade D perforation at the left T9 pedicle and transitional MT/L zone. The second 3D scan enabled screw re-insertion, correcting the perforation to grade A (Figure 2).

## 4. Discussion

The present study aimed to investigate PS insertion accuracy and perforation characteristics with the use of preoperative versus intraoperative CT navigation. Our results revealed that the accuracy of intraoperative CT navigation (97%) was significantly higher than that of intraoperative CT navigation (95%). Perforation was more than twice as likely to occur among Lenke type 1 AIS cases in the preoperative CT group. Although statistically significant, the absolute difference in accuracy between groups was modest (approximately 2%) and should be interpreted with caution. The practical benefit may lie more in the prevention of critical breaches in anatomically vulnerable zones than in the overall rate alone. Our results also demonstrated that screw perforation was more likely to occur at the concave side of PT to the MT/L transition, in which cases the use of intraoperative 3D CT navigation facilitated screw re-insertion.

A previous study found no significant difference in total screw accuracy for intraoperative versus preoperative CT [10]. However, that study did not re-evaluate screw position with a second intraoperative 3D scan to minimize radiation exposure. On the contrary, screw re-insertion during a second intraoperative 3D scan increased the placement accuracy rate [19]. In this respect, the present study reflected the importance of re-evaluating inserted screws using a second 3D scan in the intraoperative CT group. Indeed, without the second intraoperative 3D scan, the number of the grade D perforations would have been almost identical between groups. Our result suggested that use of the O-arm surely contributed to increased accuracy of AIS surgery (Table 4). Furthermore, the present study also precisely compared the two groups. Use of the second intraoperative 3D scan resulted in extra radiation exposure in the intraoperative group (Table 5).

With the second intraoperative 3D CT, the DLP was 365 ± 172 mGy-cm, an acceptable value considering national diagnostic reference levels in Japan [20]. Therefore, use of the O-arm including the second intraoperative 3D scan would be acceptable because it is consistent with the principle of medical radiation exposure benefits outweighing risks [21]. Intraoperative CT navigation was conducted using a pediatric low-dose protocol, yielding a mean DLP within national guidelines. While this adds radiation exposure compared with freehand techniques, it allows for the immediate correction of mispositioned screws, potentially avoiding the need for revision surgeries, which would otherwise result in additional radiation and operative risk.

From the literature, several factors concerning the concave side of PT are vulnerable to PS perforation. An analysis of thoracic pedicle width using normal specimens demonstrated minimal width in the T3–T5 region, especially in the T4 right pedicle [22]. However, when determining range of fixation in AIS among Lenke type 1 and 2 AIS cases, the area around T2–T4 is subjected to upper instrumented vertebra [23]. The pedicle on the concave side of the PT curve in AIS has morphological characteristics that make it riskier for PS insertion, as well as a small diameter at the isthmus [24]. Distance from the reference frame is associated with PS perforation. Oba et al. analyzed PS perforation in AIS surgery using O-arm navigation [25]. They clarified that perforation occurred in 2.7% of cases and that the perforation rate by the ninth or subsequent screws was significantly higher than for the previous screws. Chan et al. revealed that the highest rate of lateral perforation was at the concave side of the PT zone using the freehand or fluoroscopic technique [14]. At the proximal site, a screw insertion maneuver may be influenced by an inadequate skin incision site. Similar to these studies, we demonstrated that lateral perforation most frequently occurred at the concave side of the PT zone in both groups (Table 3). We also demonstrated that the confirmational intraoperative CT enabled the re-insertion of the perforated screw, thereby increasing placement accuracy compared with preoperative CT navigation (Table 4). These results highlight the need to pay attention to the area at the concave side of the PT zone when using preoperative or intraoperative CT navigation.

Notably, screw perforation more frequently occurred in the transitional PT/MT or transitional MT/L zone than at the apical vertebra, such as the concave MT, convex MT, concave L, or convex L zones, which involved the greatest degree of rotation (Table 4). On the contrary, Zhang et al. compared intraoperative and preoperative navigation techniques [10], pointing out that the apical vertebrae were most vulnerable to perforation due to being the most rotated. A possible explanation for this might be that surgeons would be more influenced by the change in trajectory at the transitional versus apical vertebrae [26].

This study has some limitations. First, screw misplacement possibly occurred during the correction maneuvers. Even if accurate screw insertion is achieved, correction forces, such as distraction, compression, and rod rotation force, can induce PS perforation in fragile bony cases [27]. Second, in the intraoperative CT group, we could not detect the exact number of the re-inserted screws because the surgeon decided whether to perform the second 3D scan. Therefore, if the second 3D scan was performed in all cases, a higher accuracy rate would have been noted in the intraoperative group. Third, DLP values were not available for the preoperative CT scans, as these were often performed externally with varying protocols and without consistent dose documentation. Therefore, a direct comparison of radiation exposure between preoperative and intraoperative CT groups was not possible. Despite these limitations, the present study results provide useful information regarding the characteristics of PS perforation using pre- versus intraoperative CT navigation in AIS surgery.

In conclusion, intraoperative CT navigation with a confirmatory second 3D scan can enhance pedicle screw placement accuracy with acceptable radiation exposure when performed using a pediatric low-dose protocol. Surgeons should carefully balance the benefits of improved accuracy with the imperative to minimize radiation exposure in adolescent patients.

## Figures and Tables

**Figure 1 jcm-14-04729-f001:**
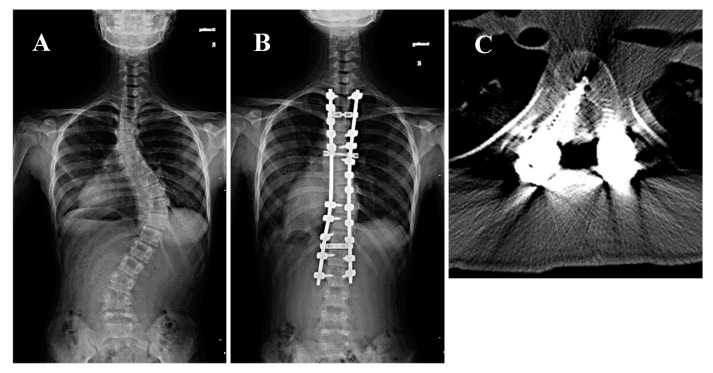
A 14-year-old girl presented with Lenke type 2A adolescent idiopathic scoliosis and a major Cobb angle of 56° at T8–L1 underwent corrective fusion surgery of T2–L2 using preoperative CT navigation. (**A**,**B**) Pre- and postoperative whole spine radiographs. The white arrow indicates the right T6 pedicle. (**C**) Postoperative CT scan showing a grade D medial perforation at the left T6 pedicle.

**Figure 2 jcm-14-04729-f002:**
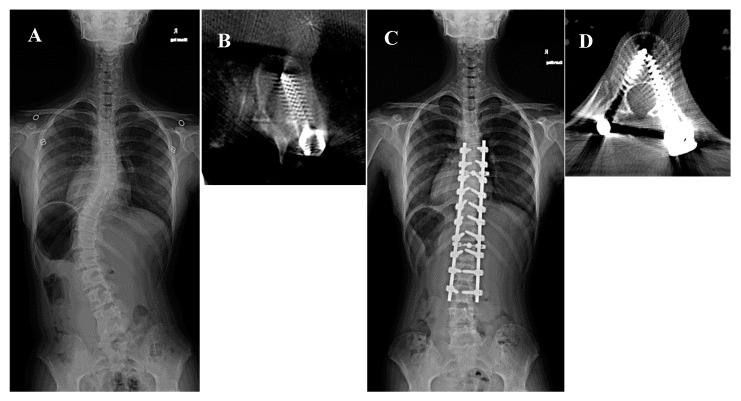
A 14-year-old girl presented with Lenke type 5C adolescent idiopathic scoliosis and a major Cobb angle of 47° at T10–L3 underwent corrective fusion surgery at T6–L3 using intraoperative computed tomography (CT) navigation. (**A**) Preoperative whole spine radiograph. The white arrow indicates the left T9 pedicle. (**B**) Intraoperative second 3D scan showing a medial grade D perforation at the left T9 pedicle. We then re-inserted the screw. (**C**) Postoperative whole spine radiograph. The white arrowhead indicates the left T9 pedicle. (**D**) Postoperative CT scan showing a grade A perforation at the left T9 pedicle.

**Table 1 jcm-14-04729-t001:** Patient demographic, curve classification and major Cobb angle.

	Preoperative CT	Intraoperative CT	*p*
N	48	59	
Age (years)	14.8 ± 1.8	14.2 ± 1.8	0.100
Female (%)	43 (86)	53 (90)	0.967
Total number of screws	853	1095	
Lenke classification			
1	26 (54)	29 (49)	0.194
2	7 (15)	13 (22)	
3	0 (0)	3 (5)	
5	11 (30)	11 (19)	
6	4 (8)	3 (5)	
Major Cobb angle (°)	47 ± 6	49 ± 8	0.082

**Table 2 jcm-14-04729-t002:** Number of screws inserted and perforation rate stratified by curve classification. Lenke type 4 curves were not present in this cohort. *; significant difference.

	Preoperative CT		Intraoperative CT		
Type	Screw (%)	Perforation (%)	Screw (%)	Perforation (%)	*p*
1	464 (54)	24 (5)	532 (49)	12 (2)	0.014 *
2	154 (18)	11 (7)	276 (26)	10 (4)	0.105
3	-	-	53 (5)	6 (11)	-
5	146 (17)	6 (4)	167 (15)	3 (2)	0.222
6	89 (10)	3 (3)	67 (6)	0 (0)	0.129
	853	44 (5)	1095	31 (3)	0.0081 *

**Table 3 jcm-14-04729-t003:** Zone distribution of the grade D perforation in preoperative and intraoperative groups. Intraoperative CT *****: Before performing second 3-dimensional scan; Intraoperative CT †: After performing second 3-dimensional scan.

	Preoperative CT		Intraoperative CT *		Intraoperative CT †	
Zone	Medial	Lateral	Medial	Lateral	medial	Lateral
Concave PT	3	5	1	3	0	1
Convex PT	2	0	0	2	0	0
Transitional PT/MT	1	0	1	1	0	1
Concave MT	0	1	0	2	0	0
Transitional MT/L	0	4	1	1	0	0
Concave L	0	2	0	1	0	1
Total	6	12	3	10	0	3

**Table 4 jcm-14-04729-t004:** Perforation grading before and after second 3-dimensional scan in the intraoperative CT group.

Type	Zone	Before	After
Lenke 1	Concave PT	C	A
Lenke 1	Concave PT	D	skip
Lenke 1	Concave PT	D	B
Lenke 2	Convex PT	D	B
Lenke 2	Convex PT	D	B
Lenke 2	Transitional PT/MT	C	A
Lenke 2	Transitional MT/L	C	A
Lenke 5	Transitional MT/L	D	B
Lenke 5	Concave MT	D	A
Lenke 6	Concave MT	D	Skip

**Table 5 jcm-14-04729-t005:** Comparison for operative time and absorbed radiation in both groups.

	Preoperative CT	Intraoperative CT	
	48	59	*p*
Operative time (min)	236 ± 48	255 ± 45	0.033 *
EBL (mL)	364 ± 267	453 ± 409	0.100
DLP (mGy-cm)	-	365 ± 172	

EBL: Estimated blood loss; DLP: dose length product; *: A significant difference.

## Data Availability

The raw data supporting the conclusions of this article will be made available by the authors on request.

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
