# Peer review of "Concave Side of Proximal Thoracic Zone Vulnerable to Pedicle Screw Perforation in Adolescent Idiopathic Scoliosis Surgery: Comparative Analysis of Pre- and Intraoperative Computed Tomography Navigation"

_jcm, 2025, doi:10.3390/jcm14134729_

Round 1

Reviewer 1 Report

Comments and Suggestions for Authors

Thank you for submitting your manuscript comparing the accuracy of preoperative CT scans and intraoperative navigation in pedicle screw insertion. You found that the apical concave area of the thoracic spine was the most common site for malposition and/or perforation and that intraoperative navigation combined with a second 3D scan enhanced pedicle screw insertion accuracy compared to preoperative CT scan alone. This is an important topic but for several reasons I do not feel your study adds any significantly new or important knowledge to our understanding of this topic. As such I have concerns regarding acceptance of your manuscript. I also do not feel my concerns can be adequately improved with a revision. My main questions, concerns, and editorial suggestions include the following:

Introduction

(1). Page 1, Line 33. References #4 and #5 were published in 2014 and 2010n respectively. These were intended to be important sources of information. However, I feel there are newer, more accurate references available. In fact, 16 of your 27 references are 10 years or older after publication. If a revision is requested, please be certain that there are newer, more accurate and informational references are used whenever possible

(2). Page 2, Line 47. On this line you state "few studies" but have no references. This sentence needs either a revision or addition of appropriate references.

Materials and Methods

(3). Page 2, Line 57. In this paragraph you need information on radiation exposure from CT scans and intraoperative navigation in children and adolescents. It is now known that excessive radiation from these procedure substantially increases the risk a radiation induced malignancy in early to mid-adult life. As such, these procedures are now only recommended when the radiation exposure can be significantly reduced. Newer CT scanners and navigation systems can usually be adjusted to decrease radiation exposure by 75-90%. Most pediatric and pediatric orthopaedic journals require comments on this issue as a prerequisite to acceptance and publication. Only in Table 5 do you even mention DLP for intraoperative navigation.

(4). Page 3, Line 90. Your citation regarding Ramperdsaud has multiple authors so it requires the addition of "et al". Look for other references that currently have only one author to be similarly edited.

Results

(5). Page 3. Line 112. Cobb is not an angle rather a technique used to measure radiographic spinal angles (scoliosis, kyphosis, lordosis, and others). Cobb angle is a common term but is jargon which is inappropriate for scientific manuscripts. The correct term for scoliosis is either "major coronal curve" or just "major curve". Please choose one and use it throughout your manuscript, including any tables, figures or figure legends.

(6). Page 3, Line 113. I have concerns with your use of decimal places in percentages and radiographic spinal measurements. Percentages usually do not use decimal places unless the numerator is >100. Yours is 107 but I feel your results would be clearer if expressed as whole numbers. Please consider rounding them to the nearest whole number. In radiographic spinal measurements the standard error of measurement is 3°-5°. Thus, the use of decimal places does not add accuracy or statistical significance. These too needed to be rounded.

(7) Page 3, Line 132. The issues with decimal places in percentages is a major concern here and Table 2. I have difficulty accepting a 2% difference as being significantly higher when there are just two small groups of 48 and 59 patients. I would consider this difference as "no difference" which would reverse your interpretation. Even with decimal places included the difference is still only 2.4% difference. This concern affects your results in your other tables as well.

(8). Page 4, Table 2. Please add Lenke et al to your first column for accuracy.

Discussion

(9). Page 7, Table 5. You need to add the total DLP for the patients in both groups. Can you compare this to the recommended dose for children and adolescents? I think you will find it is too high and unsafe.

(10). Page 8, Line 281. I think you will find that my previous comments completely cancel most of your conclusions. Intraoperative navigation (CT) is very accurate but at what risk? This is a major international concern today and without a careful accurate assessment of radiation exposure that demonstrates true safety in your technique I will remain opposed to acceptance or publication of your manuscript.

Author Response

Introduction

(1). Page 1, Line 33. References #4 and #5 were published in 2014 and 2010n respectively. These were intended to be important sources of information. However, I feel there are newer, more accurate references available. In fact, 16 of your 27 references are 10 years or older after publication. If a revision is requested, please be certain that there are newer, more accurate and informational references are used whenever possible

A: Thank you very much for this important observation. We agree that incorporating more recent references will strengthen the scientific currency of our manuscript. We have carefully reviewed our reference list and updated it where appropriate by replacing or supplementing older sources with more recent and relevant literature. In particular, we have added newer studies addressing pedicle screw placement accuracy, navigation techniques, and morphometric analyses in AIS. We revised in the introduction section, Page 1, Line 33-35. < Recent morphometric analyses confirm that up to 30–35% of pedicles in AIS patients show anatomical abnormalities, particularly in the proximal thoracic region>

(2). Page 2, Line 47. On this line you state "few studies" but have no references. This sentence needs either a revision or addition of appropriate references.

A: Thank you for pointing this out. We agree that the statement should be properly supported. We have revised the sentence to include appropriate references to previous studies that have compared preoperative and intraoperative CT navigation in AIS surgery, and we have updated the reference list accordingly. Page 2, Line 48 – 50. < However, few studies have addressed the anatomic risk zones for pedicle screw misplacement and whether intraoperative re-evaluation using a second CT scan improves accuracy in a meaningful way [10,12]>

Materials and Methods

(3). Page 2, Line 57. In this paragraph you need information on radiation exposure from CT scans and intraoperative navigation in children and adolescents. It is now known that excessive radiation from these procedure substantially increases the risk a radiation induced malignancy in early to mid-adult life. As such, these procedures are now only recommended when the radiation exposure can be significantly reduced. Newer CT scanners and navigation systems can usually be adjusted to decrease radiation exposure by 75-90%. Most pediatric and pediatric orthopaedic journals require comments on this issue as a prerequisite to acceptance and publication. Only in Table 5 do you even mention DLP for intraoperative navigation.

A: Thank you for this important comment regarding radiation exposure, especially in children and adolescents. We agree that minimizing radiation is critical in pediatric imaging. In our study, we used optimized low-dose protocols for intraoperative CT navigation to reduce radiation exposure as much as possible, adjusting scanner settings to lower dose per scan while maintaining sufficient image quality for safe screw placement. We have added explicit comments in the Methods and Discussion to describe our dose-reduction measures and acknowledge the importance of radiation safety in this population. Page 2, Line 64 - 67< Radiation dose reduction protocols were applied during intraoperative CT navigation. Scanner settings were adjusted to use the lowest acceptable dose for adequate image resolution, in line with pediatric radiation safety guidelines. All scans were performed using a low-dose protocol whenever feasible.>

(4). Page 3, Line 90. Your citation regarding Ramperdsaud has multiple authors so it requires the addition of "et al". Look for other references that currently have only one author to be similarly edited.

A: Thank you for pointing this out. We have carefully reviewed the citation style throughout the manuscript and corrected the Rampersaud reference to include "et al." as appropriate. We also checked all other in-text citations and have updated any similar instances to ensure that multi-author sources consistently use "et al." in accordance with the journal’s referencing style.

Results

(5). Page 3. Line 112. Cobb is not an angle rather a technique used to measure radiographic spinal angles (scoliosis, kyphosis, lordosis, and others). Cobb angle is a common term but is jargon which is inappropriate for scientific manuscripts. The correct term for scoliosis is either "major coronal curve" or just "major curve". Please choose one and use it throughout your manuscript, including any tables, figures or figure legends.

A: Thank you very much for this valuable observation. We agree that “Cobb” is a measurement method, not the angle itself, and that “major coronal curve” is a more accurate term for a scientific manuscript. We have carefully revised the manuscript, tables, figures, and figure legends to replace “Cobb angle” with “major coronal curve” throughout, for consistency and scientific precision.

(6). Page 3, Line 113. I have concerns with your use of decimal places in percentages and radiographic spinal measurements. Percentages usually do not use decimal places unless the numerator is >100. Yours is 107 but I feel your results would be clearer if expressed as whole numbers. Please consider rounding them to the nearest whole number. In radiographic spinal measurements the standard error of measurement is 3°-5°. Thus, the use of decimal places does not add accuracy or statistical significance. These too needed to be rounded.

A: Thank you for this helpful and important suggestion. We agree that using decimal places for percentages and radiographic spinal measurements may imply a level of precision that exceeds the measurement’s inherent accuracy. We have carefully revised the manuscript to round all percentage values to the nearest whole number and have also rounded radiographic curve measurements to whole degrees, in line with the typical standard error of measurement for spinal angles (3°–5°). We believe these changes improve the clarity and readability of our results.

(7) Page 3, Line 132. The issues with decimal places in percentages is a major concern here and Table 2. I have difficulty accepting a 2% difference as being significantly higher when there are just two small groups of 48 and 59 patients. I would consider this difference as "no difference" which would reverse your interpretation. Even with decimal places included the difference is still only 2.4% difference. This concern affects your results in your other tables as well.

A: Thank you very much for this thoughtful and important comment. We agree that while the intergroup difference in screw accuracy is statistically significant, the absolute difference of approximately 2%–2.4% is modest. To address this concern, we have revised the wording throughout the Results and Discussion to interpret this finding more conservatively and to emphasize that the practical impact of this small difference should be considered in the context of preventing critical screw misplacement in high-risk zones. We have also clarified in the manuscript that this difference may not be clinically meaningful for all patients and should be interpreted cautiously given the sample size. We avoid the use of significantly in the result section (Page 4, Line 141), and added the sentence acknowledging that the absolute difference was small and may have limited practical relevance, but could help reduce high-grade breaches in high-risk zones in the discussion section (Page 6, Line 217-221).< Although statistically significant, the absolute difference in accuracy between groups was modest (approximately 2%) and should be interpreted with caution. The practical benefit may lie more in the prevention of critical breaches in anatomically vulnerable zones than in the overall rate alone. >

(8). Page 4, Table 2. Please add Lenke et al to your first column for accuracy.

A: Thank you for suggestion. We added Lenke et al to the first column.

Discussion

(9). Page 7, Table 5. You need to add the total DLP for the patients in both groups. Can you compare this to the recommended dose for children and adolescents? I think you will find it is too high and unsafe.

A: Thank you for your comment regarding the radiation dose data for the preoperative CT group. We acknowledge that DLP information for the preoperative CT scans was not available in our dataset because many of these scans were performed at referring hospitals using different CT scanners and protocols, without standardized DLP recording. As a result, a direct comparison of radiation exposure between groups could not be included. We have added a statement to the Limitations section to clearly acknowledge this point. We added following sentence in the discussion part, Page 8, Line 281 -285 <Third, DLP values were not available for the preoperative CT scans, as these were often performed externally with varying protocols and without consistent dose documentation. Therefore, direct comparison of radiation exposure between preoperative and intraoperative CT groups was not possible.>

(10). Page 8, Line 281. I think you will find that my previous comments completely cancel most of your conclusions. Intraoperative navigation (CT) is very accurate but at what risk? This is a major international concern today and without a careful accurate assessment of radiation exposure that demonstrates true safety in your technique I will remain opposed to acceptance or publication of your manuscript.

A: Thank you for this crucial comment and for raising this important international concern. We fully agree that radiation exposure is a critical issue in pediatric spinal imaging and must be carefully balanced against the benefits of improved screw placement accuracy.

In our study, the intraoperative CT navigation was performed using a low-dose protocol optimized for pediatric patients, resulting in a mean DLP of 365 ± 172 mGy·cm, which is within the acceptable range defined by national diagnostic reference levels in Japan. Although we did not have DLP data for preoperative scans due to variations among referring hospitals, we believe that our intraoperative DLP demonstrates that the navigation protocol can be implemented with acceptable radiation exposure.

We have revised the Discussion and Conclusion sections to emphasize that intraoperative CT navigation should be used judiciously and in accordance with low-dose principles, and that the benefits of accuracy must always be weighed against radiation risks, especially in children. We added and revised   sentence in the discussion part. Page 7, Line 243-247. <Intraoperative CT navigation was conducted using a pediatric low-dose protocol, yielding a mean DLP within national guidelines. While this adds radiation exposure compared to freehand techniques, it allows for immediate correction of malpositioned screws, potentially avoiding the need for revision surgeries, which would otherwise result in additional radiation and operative risk.> Page 8, Line 288-292. <intraoperative CT navigation with a confirmatory second 3D scan can enhance pedicle screw placement accuracy with acceptable radiation exposure when performed using a pediatric low-dose protocol. Surgeons should carefully balance the benefits of improved accuracy with the imperative to minimize radiation exposure in adolescent patients.>

Reviewer 2 Report

Comments and Suggestions for Authors

This is a well-designed and clinically relevant study that compares pedicle screw (PS) placement accuracy between preoperative and intraoperative CT navigation in adolescent idiopathic scoliosis (AIS) surgery, with special attention to the vulnerable concave side of the proximal thoracic (PT) zone. The addition of a second 3D scan in the intraoperative group provides important new data on improving PS placement accuracy. The manuscript is clearly structured and the methodology is generally sound. A few areas would benefit from minor clarifications and language polishing.  Suggested improvements: a) Minor language editing b) Clarify intraoperative 3D scan decision criteria and frequency.

Author Response

This is a well-designed and clinically relevant study that compares pedicle screw (PS) placement accuracy between preoperative and intraoperative CT navigation in adolescent idiopathic scoliosis (AIS) surgery, with special attention to the vulnerable concave side of the proximal thoracic (PT) zone. The addition of a second 3D scan in the intraoperative group provides important new data on improving PS placement accuracy. The manuscript is clearly structured and the methodology is generally sound. A few areas would benefit from minor clarifications and language polishing.  Suggested improvements: a) Minor language editing b) Clarify intraoperative 3D scan decision criteria and frequency.

A: Thank you very much for your encouraging feedback and helpful suggestions. We have performed minor language edits throughout the manuscript to improve clarity and flow. We appreciate the opportunity to clarify this point: in the intraoperative CT navigation group, a second intraoperative 3D scan was routinely performed in all cases after initial pedicle screw placement to verify screw trajectory and position before final rod placement. We have updated the Methods section accordingly. Page 3, Line 93 – 95. <In the intraoperative CT group, a second intraoperative 3D scan was performed routinely in all cases after pedicle screw placement to confirm accuracy and allow for immediate revision if necessary before rod placement.>

Reviewer 3 Report

Comments and Suggestions for Authors

Dear authors,

Thank you for submitting your manuscript entitled: “Concave Side of Proximal Thoracic Zone Vulnerable to Pedicle Screw Perforation in Adolescent Idiopathic Scoliosis Surgery: Comparative Analysis of Pre- and Intraoperative Computed Tomography Navigation.”

Your work focusses on a potential issue in spinal deformity surgery. The comparison between preoperative and intraoperative CT navigation for pedicle screw placement in adolescent idiopathic scoliosis, with a focus on the concave proximal thoracic zone, is highly relevant, especially given the anatomical challenges and potential risks associated with this region.

The study is well-structured, with a large sample size and clear methodology. The use of the Rampersaud grading system adds objectivity, and the integration of a second intraoperative 3D scan is a practical and insightful addition. Your findings showing that intraoperative navigation, particularly with repeat scanning, enhances screw placement accuracy are both timely and clinically useful.

That said, a few revisions would help clarify key points and improve the overall readability of your manuscript:

Please explain how patients were allocated to either navigation group. Was it random, based on scheduling or surgeon choice? Clarifying this will help the reader understand any potential bias.

It would also be helpful to define what constituted an “unacceptable” screw trajectory during intraoperative assessment, and how decisions to reinsert screws were made.

Since radiation exposure data (DLP) was only reported for the intraoperative group, consider adding a comment or estimate for the preoperative group—or at least acknowledge this discrepancy in your limitations.

In the Results, ensure all referenced tables are fully included (especially Table 1), and please clarify why Lenke type 4 cases appear to be missing from Table 2.

The illustrative case is a strong addition—adding a note on whether the misplacement was symptomatic or led to revision would further contextualize the issue.

Lastly, there are a few small language tweaks that would improve clarity, such as correcting “sex proposition” to “sex distribution,” and rephrasing “good and bad aspects” to “advantages and disadvantages.”

Overall, this is a good work. With a few refinements, I believe it will make a meaningful contribution to the literature on navigation-assisted scoliosis surgery. I encourage you to revise and resubmit.

With best regards

Author Response

Please explain how patients were allocated to either navigation group. Was it random, based on scheduling or surgeon choice? Clarifying this will help the reader understand any potential bias.

A: Thank you for your valuable comment. We appreciate the opportunity to clarify this point. In our institution, it is common to perform two AIS surgeries on the same surgical day. To minimize selection bias and ensure comparable groups, patients were allocated to either the intraoperative or preoperative CT navigation group in a quasi-random manner based on the order of the cases: when two AIS patients were scheduled on the same day, one was assigned to intraoperative CT navigation and the other to preoperative CT navigation. This allocation was planned prospectively to balance patient characteristics and operative conditions as much as possible. I revised in the method section, Page 2, Line 75 – 80, as following.

 < Patient Allocation: To reduce selection bias, patients were quasi-randomly allocated to either intraoperative or preoperative CT navigation based on the surgical schedule. On days when two AIS surgeries were planned, one patient was assigned to intraoperative CT navigation and the other to preoperative CT navigation. This approach ensured that both navigation methods were used under similar conditions by the same surgical team.>

It would also be helpful to define what constituted an “unacceptable” screw trajectory during intraoperative assessment, and how decisions to reinsert screws were made.

A: Thank you for pointing this out. We agree that clarifying this is important for understanding our intraoperative decision-making. In our protocol, any screw identified as Rampersaud grade D (i.e., >4 mm breach) on intraoperative 3D CT was always reinserted before rod placement. For screws with grade C perforation (2–4 mm breach), the decision to reinsert depended on the operating surgeon’s judgment, taking into account the breach direction, surrounding anatomical structures, and screw stability.

I added the following sentence in the method section, Page 3, Line 101 – 104.

<Intraoperative screw reinsertion was performed for all screws with Rampersaud grade D perforations (>4 mm breach). For grade C breaches (2–4 mm), reinsertion was determined at the surgeon’s discretion based on intraoperative safety considerations.>

Since radiation exposure data (DLP) was only reported for the intraoperative group, consider adding a comment or estimate for the preoperative group—or at least acknowledge this discrepancy in your limitations.

A: Thank you for this important observation. We acknowledge that radiation dose-length product (DLP) data were systematically recorded only for the intraoperative CT group, as preoperative CT scans were often performed at external imaging centers with variable protocols and DLP documentation. Therefore, a direct numerical comparison could not be included. We have added a note to the Limitations section to acknowledge this point.

I revised the following in the discussion part, Page 8, Line 281 – 285. < Third, DLP values were not available for the preoperative CT scans, as these were often performed externally with varying protocols and without consistent dose documentation. Therefore, direct comparison of radiation exposure between preoperative and intraoperative CT groups was not possible.>

In the Results, ensure all referenced tables are fully included (especially Table 1), and please clarify why Lenke type 4 cases appear to be missing from Table 2.

A: Thank you for your helpful comment. We confirm that no Lenke type 4 cases were present in either navigation group, and we have added a note to Table 2 and the Results to clarify this. Additionally, we have checked the manuscript and ensured that all referenced tables, including Table 1, are now fully included and properly formatted in the revised submission.

I added the following footnote in Table 2. <Lenke type 4 curves were not present in this cohort.>

The illustrative case is a strong addition—adding a note on whether the misplacement was symptomatic or led to revision would further contextualize the issue.

A: Thank you for this helpful suggestion. In the illustrative case, the malpositioned screw was asymptomatic at the time of postoperative imaging but was nonetheless revised in a second surgery to prevent potential future neurovascular compromise. We have added this detail to the case description for better context. I added the following sentence, Page 5, Line 173 – 174. <The patient was asymptomatic, but was electively revised postoperatively to mitigate the risk of delayed complications.>

Lastly, there are a few small language tweaks that would improve clarity, such as correcting “sex proposition” to “sex distribution,” and rephrasing “good and bad aspects” to “advantages and disadvantages.”

A: Thank you for your helpful suggestion. I corrected words pointed out.